# Comparison of Surface Properties of Sepiolite and Palygorskite: Surface Energy and Nanoroughness

**DOI:** 10.3390/nano11061579

**Published:** 2021-06-16

**Authors:** Ricardo Almeida, Eduardo Ferraz, Julio Santarén, José A. F. Gamelas

**Affiliations:** 1CIEPQPF—Chemical Process Engineering and Forest Products Research Centre, Department of Chemical Engineering, University of Coimbra, Rua Sílvio Lima, Pólo II, PT-3030-790 Coimbra, Portugal; ricardsalmeida00@gmail.com; 2Techn&Art, Polytechnic Institute of Tomar, Quinta do Contador, Estrada da Serra, PT-2300-313 Tomar, Portugal; ejmoferraz@ipt.pt; 3Geobiotec, Geosciences Department, Campus Universitário de Santiago, University of Aveiro, PT-3810-193 Aveiro, Portugal; 4Tolsa, SA, Research & Technology for New Businesses, Ctra. de Madrid a Rivas Jarama, 35, ES-28031 Madrid, Spain; jsantaren@tolsa.com

**Keywords:** adsorption, aluminosilicate, attapulgite, clay minerals, fibrous minerals, inverse gas chromatography

## Abstract

The surface properties of two sepiolite samples and one palygorskite sample were compared using inverse gas chromatography (IGC). Samples were previously conditioned at appropriate temperatures for the removal of all zeolitic water. Dispersive (or Lifshitz–van der Waals) component of the surface energy (γ_s_^d^), specific interactions (−ΔG_a_^s^) with π electron donor bases (1-alkenes), and nanomorphology indices ${(\mathrm{IM}\chi }_{T})$ based on the injections of cycloalkanes and a branched alkane were measured. From IGC data, at 240 °C, it was found that the palygorskite was clearly distinguished from the sepiolites. The palygorskite possessed a lower γ_s_^d^, larger −ΔG_a_^s^ with 1-alkenes, and remarkably higher ${\mathrm{IM}\chi }_{T}$. Slight differences could also be observed between the two sepiolite samples with the same origin. The results were rationalized in terms of the structural features of the two studied minerals. The larger channels of the sepiolite allow for a better insertion of the *n*-alkanes (longer retention times) while excluding the bulkier probes, such as cyclooctane or 2,2,4-trimethylpentane. Accordingly, the corresponding γ_s_^d^ values were larger and the ${\mathrm{IM}\chi }_{T}$ values were lower (higher surface nanoroughness) for the sepiolites. Regarding Lewis acid–base properties, all the sample’s surfaces evidenced a very strong amphoteric character. The present results highlight the potential of the evaluated samples for, e.g., adsorption processes with volatile organic compounds or matrix–filler interactions regarding the production of composite structures with Lewis acid–base matrices.

## 1. Introduction

Sepiolite and palygorskite are fibrous clay minerals composed of discontinuous structural blocks, alternating with structural tunnels. Each block has a 2:1 phyllosilicate structure (two silica tetrahedral sheets with a central octahedral sheet) and is linked to the next inverted tetrahedral sheet by Si–O–Si bonds [1,2]. Sepiolite is a trioctahedral mineral with the ideal formula Si_12_O_30_Mg_8_(OH)_4_(OH_2_)_4_·8H_2_O, having all the octahedral sites occupied by magnesium. In contrast, palygorskite, with the ideal formula Si_8_O_20_(Al_2_Mg_2_)(OH)_2_(OH_2_)_4_·4H_2_O, has a dioctahedral character, and only four of the five octahedral sites are occupied, mainly with magnesium and aluminum ions [2,3]. The structural tunnels of these two minerals have cross-section dimensions of 1.06 × 0.37 nm^2^ and 0.64 × 0.37 nm^2^ for sepiolite and palygorskite, respectively, and inside them, there are exchangeable cations and zeolitic water. When the zeolitic water of these tunnels is removed by heating, the sorption capacity is increased and small organic molecules could also be adsorbed [2]. Due to the special sorptive, colloidal–rheological, and catalytic properties of these two minerals, they have attracted a growing interest for a wide range of applications [4,5].

Inverse gas chromatography (IGC) is a powerful technique for the characterization of a material’s surface properties. This tool can be used to study a wide variety of materials including polymers, cellulosic materials, minerals and inorganic materials, pharmaceuticals and medical products, supported catalysts, and microporous materials [6,7,8]. IGC is a variation of conventional gas chromatography (GC), and the main distinguishing factor between both techniques is related to the mobile and stationary phases. In IGC, the solid material under investigation (stationary phase) is packed into the column and a volatile probe molecule (single gas or vapor) of known characteristics is injected into the flow of an inert carrier gas and passes through the column [7,8]. From the determined retention times of different vapor probes on the surface of the solid material, a wide range of physicochemical properties can be obtained, including dispersive component of the surface energy of the material, specific components of the free energy, enthalpy and entropy of adsorption of the probes, nanoroughness, Flory–Huggins parameters, and glass transition temperatures. IGC is a very accurate and versatile characterization tool. In particular, IGC can overcome the contact angle measurement limitations in the analysis of porous, rough, and heterogeneous surfaces [6,7,9].

Some studies have already been carried out to investigate the surface properties of sepiolite and palygorskite by IGC.

Morales et al. (1991) used IGC analysis to determine the enthalpy and entropy of adsorption of *n*-alkanes (*n* = 6, 8, and 9) and of two polar probes (dichloromethane and furane) and the dispersive component of the surface energy (γ_s_^d^) and to assess briefly the Lewis acid–base characteristics of a sepiolite sample pretreated with hydrogen peroxide. It was observed in the range of evaluated temperatures (100–130 °C) that the γ_s_^d^ of sepiolite increased with increasing temperature, varying from 54 to 74 mJ m^−2^. Based on the measured acidity and basicity constants (K_A_ and K_D_, respectively), the authors proposed that the sample surface was slightly more acidic than basic [10]. A few years later, Așkin and Yazici (2005) estimated several surface properties of another sepiolite sample in a quite different range of temperatures (320–350 °C) by measuring the retention times of different nonpolar and polar probes. The results showed that γ_s_^d^ values decreased with increasing temperature (320–350 °C), ranging from 84 (320 °C) to 71 mJ m^−2^ (350 °C). These authors also made a comparison with other clay materials concerning the dispersive component of the surface energy and concluded that γ_s_^d^ values for sepiolite were rather lower than those obtained for kaolinite, illite, or bentonite. Their studies also pointed to a prevalent acidic character of the sepiolite surface, under the measured conditions [11]. Lazarevic et al. (2009) [12] studied the adsorption properties of sepiolite by IGC at zero and finite surface coverage between 210 and 240 °C. From IGC data at zero coverage, they found that dispersive component values decreased with temperature increase, ranging from 150 (210 °C) to 130 mJ m^−2^ (240 °C). However, based on the calculation of the K_A_ and K_D_ constants, they proposed that sepiolite surface had a prevalent basic character, in contrast to what was concluded in the other previous works [10,11].

Boudriche et al. (2010) evaluated the influence of a grinding process on the surface properties of palygorskite by IGC. The dispersive component at 130 °C, before grinding, was found to be 164 mJ m^−2^. With the grinding, the γ_s_^d^ values decreased slightly (down to 143 mJ m^−2^). From the calculation of the nanomorphology index $\left({\mathrm{IM}\chi }_{T}\right)$ of 2,3,4-trimethylpentane and cyclooctane, the authors demonstrated that the surface morphology of palygorskite was not significantly influenced by the grinding process [13].

The aim of the present work was to make a comparative study of two sepiolite samples and one palygorskite sample (provided by TOLSA), measured in the same temperature conditions and after an appropriate preconditioning. These samples were thoroughly analyzed by IGC, which included the determination of the dispersive (Lifshitz–van der Waals) component of the surface energy (γ_s_^d^), specific component of the probes’ free energy of adsorption (−ΔG_a_^s^), and surface nanomorphology indices (${\mathrm{IM}\chi }_{T}$), by injecting a wide range of probes (*n*-alkanes, 1-alkenes, cycloalkanes, and a branched alkane). The obtained results could be of interest for optimizing adsorption processes using this clay mineral type or valuable in the area of the production of composites, where the determined parameters can be used to predict compatibility between the clay surface and other components.

## 2. Materials and Methods

### 2.1. Clays

The geogenic sepiolite, supplied by Tolsa, SA (Madrid, Spain), was taken from the deposit of Vallecas-Vicálvaro (Madrid, Spain). The raw material was submitted to dry micronization using a jet mill to break the fiber bundles down into micron-size particles (sepiolite 1 (Sep. 1)) or to wet micronization producing an extensive deagglomeration of the bundles without affecting their aspect ratio (sepiolite 2 (Sep. 2)).

The geogenic palygorskite (Pal) was extracted from a deposit located in the department of M’bour, region of Thiès, south of Dakar (Senegal), and was also supplied by Tolsa, SA (Madrid, Spain). The raw material was preprocessed by micronization in a roller mill.

The general characterization of these fibrous clay samples was reported in detail elsewhere [14,15].

Powder X-ray diffractograms were obtained using a Philips X’Pert MPD diffractometer (Amsterdam, the Netherlands), with CoKα radiation (λ = 1.7903 Å). Diffractograms were collected by the counting method (step 0.025° and time 1.0 s) in the 2θ range of 5–60°. Crystalline phases were discriminated by comparison with reference diffractograms from the International Centre for Diffraction Data.

Thermogravimetric plots were acquired using a TA Instruments simultaneous TGA-DSC thermal analyzer, model SDT Q600 (TA Instruments, New Castle, DE, USA). The samples were heated from room temperature up to 1000 °C, at a rate of 10 °C/min, under a nitrogen atmosphere.

Brunauer, Emmett, and Teller (BET) specific surface area was determined by nitrogen adsorption at 77 K using a Micromeritics Gemini V analyzer (Micromeritics Instruments, Norcross, GA, USA). The samples were previously degassed at 124 °C under nitrogen flow for 12 h.

### 2.2. Inverse Gas Chromatography (IGC)

Prior to IGC analysis, the powders of the two sepiolite samples and the palygorskite sample were compacted using a hydraulic press. The obtained pressed disc was then crumbled and sieved between 850 and 500 µm before being packed in the IGC column. This procedure allowed a sufficient gas flow rate in the IGC column to be obtained.

The IGC analysis was performed using a DANI GC 1000 digital pressure control gas chromatograph equipped with a hydrogen flame ionization detector. Stainless steel columns, 0.4 m long and 0.4 cm inside diameter, were washed with acetone, dried, and passed through compressed air before packing. For these analyses, three different columns were run for each sample: 2.3–2.7 g of sepiolite 1, 2.0–2.4 g of sepiolite 2, and 2.7–3.2 g of palygorskite were packed inside the gas chromatograph column. The packed columns were shaped to fit the detector/injector geometry of the instrument and, after that, were conditioned overnight (14 h) at 290 °C for the sepiolite samples and at 240 °C for the palygorskite sample, under a helium flow (*p* = 0.05 bar), before any measurements were made. Measurements were then carried out at 240 °C, for all samples, with injector and detector kept at 180 and 200 °C, respectively. Helium (high purity, 99.99%) was used as the carrier gas with flow rates (measured with a digital flow meter) between 55 and 92 mL min^−1^. The helium flow rate was controlled in all series of experiments. Small quantities of probe vapor (<1 μL) were injected into the carrier gas, allowing work under infinite dilution conditions. *n*-alkanes (*n*-pentane (C_5_), *n*-hexane (C_6_), *n*-heptane (C_7_), and *n*-octane (C_8_)), 1-alkenes (1-pentene and 1-hexene), cycloalkanes (cyclohexane (cyclo 6) and cyclooctane (cyclo 8)), and a branched alkane (2,2,4-trimethylpentane (2,2,4-TMP)) were the probes used for the IGC data collection. Tetrahydrofuran, diethyl ether, ethyl acetate, and acetone were also injected, but due to their strong interaction with the studied materials, they could not be eluted through the column. On the other hand, injections of trichloromethane and dichloromethane provided irreproducible signals (null or very weak), which were not further considered for the analysis. All probes used were of chromatographic grade. Methane was used as the reference probe. The retention times (t_r_) were the average of three injections and were determined by the Conder and Young method [16]. Coefficient of variation between injections was lower than 5%. From the retention time data, the dispersive component of the surface energy, the specific component of the probes’ free energy of adsorption, and the surface nanomorphology indices were determined.

### 2.3. IGC Theory

The principles and the mathematical aspects of the IGC theory in infinite dilution conditions have been widely described in the literature [6,7,8,17]. Briefly, by the Schultz and Lavielle approach, the dispersive component of the surface energy (Lifshitz–van der Waals component, in the strict sense) for the solid material under analysis may be estimated from the slope of the linear fit of RTln(V_n_) as a function of 2N.*a*(γ_l_^d^)^0.5^ obtained with the injection of apolar probes, usually *n*-alkanes (Equation (1)) [18].
(1)$$\mathrm{RTln}\left(V_{n}\right)=\sqrt{\gamma_{s}^{d}}\;2N\cdot a\sqrt{\gamma_{1}^{d}}+K$$
where R is the ideal gas constant; T is the column absolute temperature; V_n_ is the net retention volume of the probe; N is the Avogadro number; *a* is the molecular surface area of the probe, γ_s_^d^ and γ_1_^d^ are the dispersive components of the surface free energy of the interacting solid material and probe, respectively; and K is a constant that is dependent on the chosen reference state.

The dispersive component of the surface energy can also be estimated by the Dorris and Gray approach. Here, the γ_s_^d^ value is determined from the difference in the free energy of adsorption due to the introduction of an additional methylene group into the carbon chain of an *n*-alkane probe (Equation (2)) [19].
(2)$$\gamma_{s}^{d}=\frac{{\left[\mathrm{RTln}\frac{V_{n}^{{(C}_{n+1}H_{2n+4})}}{V_{n}^{{(C}_{n}H_{2n+2})}}\right]}^{2}}{{4N}^{2}{\left(a_{{\mathrm{CH}}_{2}}\right)}^{2}\gamma_{{\mathrm{CH}}_{2}}}$$
where $a_{{\mathrm{CH}}_{2}}$ is the molecular area of the methylene group (0.06 nm^2^) and $\gamma_{{\mathrm{CH}}_{2}}$ is the surface energy of a solid entirely composed of methylene groups (Equation (3)).
(3)$$\gamma_{{\mathrm{CH}}_{2}}=35.6+0.058\left(293.15-T\right)$$

For probes able to establish molecular interactions with the solid material other than dispersive ones (e.g., Lewis acid–base interactions), there is a corresponding specific component contribution, ΔG_a_^s^, to the overall free energy of adsorption, ΔG_a_ [17]. The ΔG_a_^s^ contribution can be estimated by calculating the difference between the experimental value of RTln(V_n_) obtained for the respective probe (Lewis acidic, basic, or amphoteric) and the corresponding RTln(V_n_) for the reference *n*-alkane probe (Equation (4)).
(4)$$\Delta G_{a}^{s}=-\mathrm{RTln}\frac{V_{n}}{V_{n,\mathrm{ref}}}$$

To evaluate the nanoroughness of the studied clayey materials, the nanomorphology index ${(\mathrm{IM}\chi }_{T})$ was calculated according to two different approaches. The Brendlé and Papirer approach compares the measured $\chi_{\exp}$ with the $\chi_{T}$ topological index (Equation (5)) [20]. Here, $\chi_{T}$ is the previously defined topological index of a branched alkane or cycloalkane, determined using Wiener’s indices, and represents the number of carbon atoms of a hypothetical linear alkane that would interact with the surface of an ideal solid (without surface roughness) the same way as the branched alkane or the cycloalkane. $\chi_{\exp}$ is the experimental value calculated for this topological index based on the measurement of the retention times of the nonlinear alkane and the *n*-alkanes on the solid material under analysis.
(5)$${\mathrm{IM}\chi }_{T}=100\;\times \;\frac{\left(\chi_{\exp}-\chi_{T}\right)}{\chi_{T}}$$

Typically, for less rough surfaces, $\chi_{\exp}$ approaches $\chi_{T}$ and ${\mathrm{IM}\chi }_{T}$ tends to 0%; on the contrary, the rougher the surface is, the more negative the ${\mathrm{IM}\chi }_{T}$ value is.

Balard et al. (2000) suggested another approach for ${\mathrm{IM}\chi }_{T}$ calculation. In this approach, the nanomorphology index is determined based on the distance between the representative point of the branched alkane or cycloalkane and the reference *n*-alkane straight line, ΔG_a_^M^ (Equation (6)) [21].
(6)$${\mathrm{IM}\chi }_{T}{=e}^{\frac{-\Delta G_{a}^{M}}{\mathrm{RT}}}$$

When the solid surface roughness is low, the ${\mathrm{IM}\chi }_{T}$ value tends to 1, and when the surface roughness increases, the ${\mathrm{IM}\chi }_{T}$ value decreases due to the limited access of the branched alkanes or cycloalkanes to the rough parts of the solid surface compared to the linear *n*-alkanes.

## 3. Results and Discussion

### 3.1. Sample Characterization by X-ray Diffraction and Thermogravimetry

The sepiolite samples (Sep. 1 and Sep. 2) and the palygorskite sample (Pal) were previously analyzed by X-ray diffraction to evaluate the mineral phases present. Sep. 1 and Sep. 2 showed the presence of only sepiolite; Pal was mainly composed of palygorskite and quartz, with minor amounts of Ca–Mg-smectite, apatite-rich phase, calcite, dolomite, opal-CT, and sepiolite (Figure 1).

In order to choose the most suitable conditioning temperature for the IGC analyses, a thermal analysis was previously carried out. Based on the thermogravimetry (Appendix A) and the corresponding derivative (Figure 2), it was observed that the two sepiolite samples had quite similar thermal behavior, while the palygorskite sample exhibited a differing thermal behavior. In order to allow the removal of zeolitic water, temperatures of around 140 °C, i.e., after the first step of mass loss in the thermogravimetry, were firstly selected. However, when performing the conditioning followed by the analysis at this temperature, retention times of the probes, even of the short-chain *n*-alkanes, were too long to be measured, which did not allow the analysis. Then, temperatures after the second step of mass loss were selected, which according to the values of mass loss registered would correspond to the release of all zeolitic water and more 1–2 water molecules coordinated in the structure. The conditioning temperature (Temp. cond.) was therefore 240 °C for palygorskite and 290 °C for sepiolites (Figure 2), which afforded the removal of all zeolitic water, although some fraction of coordinated water was removed as well. To compare the studied materials under identical operational conditions, the IGC analyses were performed at a measurement temperature of 240 °C.

The specific surface area was also determined, and the values of 288, 337, and 129 m^2^ g^−1^ were found for Sep. 1, Sep. 2, and Pal, respectively. A detailed and exhaustive characterization of the studied samples is reported by Alves et al. (2020) and Ferraz et al. (2021) [14,15].

### 3.2. Dispersive Component of the Surface Free Energy

The dispersive component of the surface free energy (γ_s_^d^) was determined from the retention times of C_5_ to C_8_ *n*-alkanes by the Schultz–Lavielle and Dorris–Gray methods. The obtained results are presented in Table 1.

Firstly, it can be seen in Table 1 that the γ_s_^d^ values obtained by the Dorris–Gray method are much higher than those obtained by the Schultz–Lavielle method (by approximately 50%). The comparison of γ_s_^d^ values obtained by these two methods was also conducted previously by Gamelas et al. (2014), where the authors studied the surface properties of calcined kaolinitic clays by IGC. For measurements performed at 110 °C, the Dorris–Gray method provided values ca. 20% larger than those obtained by the Schultz–Lavielle method [22]. Additionally, Gamelas and Martins (2015), when analyzing carbonated and noncarbonated hydroxyapatites, noted that when the analysis temperature was 37 °C, the difference of the results between the two methods was very small (ca. 1 mJ m^−^^2^), but when the analysis temperature increased to 100 °C, this difference increased to about 5 mJ m^−^^2^ [23]. It can be concluded that the difference in the γ_s_^d^ values obtained by the two mentioned methods is inherent to the calculation approaches employed and increases when the measurement temperature increases. In the present study, since the measurement temperature was very high (240 °C), the difference between the γ_s_^d^ values obtained by these two methods was remarkably high. Additionally, it was found that the γ_s_^d^ values of the three studied samples increased in the following order: Pal < Sep. 1 < Sep. 2. Moreover, during the injections of the *n*-alkanes, the retention times of these probes were significantly longer for sepiolite samples (for similar carrier gas flow rate), which was translated into higher dispersive component values for the sepiolites. These results, for similar measurement temperature, can be attributed to the larger dimensions of the structural tunnels and channels of sepiolite, which enable higher retention of the *n*-alkane probes. When comparing the two sepiolites, the more extensive deagglomeration of the fiber bundles in Sep. 2, with a higher specific surface area, may be the reason for a somewhat higher γ_s_^d^ value compared to Sep. 1. The γ_s_^d^ values obtained in this study for the sepiolite samples (205 and 229 mJ m^−^^2^ for Sep. 1 and Sep. 2, respectively) appear to be higher than those found in the literature [10,11,12]. In particular, the present results were higher when compared specifically with the results of Lazarević et al. (2009), who studied sepiolite by IGC, between 210 and 240 °C, and obtained a dispersive component value (by Dorris–Gray method) of 130 mJ m^−^^2^ at 240 °C [12]. Despite the fact that they used the same temperature as ours, differences in geology and pretreatment of the sepiolites can be the origin of such different results in the dispersive component. Boudriche et al. (2010) also analyzed a palygorskite sample by IGC and obtained a dispersive component value very similar to ours (164 mJ m^−^^2^ at 130 °C) [13]. However, since the operating temperature used by Boudriche and coworkers was much lower than ours, and due to differences in the pretreatment and in the mineral composition of the palygorskite samples, an accurate comparison with this result cannot be made.

### 3.3. Specific Interaction Parameters

As mentioned in Section 2.2, typical Lewis basic and amphoteric probes such as tetrahydrofuran, diethyl ether, ethyl acetate, or acetone were retained too strongly in the studied clay materials and could not be eluted through the chromatographic columns. Thus, in order to be able to measure the Lewis acidic character of the mineral surface, weak Lewis bases (1-alkenes) were injected into the IGC column. In addition to these probes, cycloalkanes (cyclohexane and cyclooctane) and a branched alkane (2,2,4-trimethylpentane) were also injected. The specific interactions of adsorption (−ΔG_a_^s^) of the injected probes on the surface of the studied materials were determined by the difference between the RTln(V_n_) values of 1-alkenes, cycloalkanes, or branched alkane and the RTln(V_n_) values of the corresponding *n*-alkanes with the same number of carbon atoms. It should be noted that, actually, for the cycloalkanes or the branched alkane, the intermolecular forces with the solid material are of similar nature to those involving the *n*-alkanes. Therefore, in these cases, the “specific” interaction does not correspond to an additional interaction to the dispersive Lifshitz–van der Waals forces but rather is a negative contribution, as demonstrated below. The results are shown in Figure 3.

The specific interactions with the π electron donor bases (1-alkenes) were significantly higher for palygorskite (near 10 kJ mol^−^^1^) than for sepiolite samples (did not exceed 4 kJ mol^−^^1^), suggesting that the surface of the palygorskite particles showed the most pronounced Lewis acidic character of the three studied samples (Figure 3). Additionally, a slight decrease in the −ΔG_a_^s^ values of 1-alkenes was observed with the increase in the carbon atom number. This trend has already been reported for other mineral materials [24]. To our knowledge, specific interactions of weak Lewis bases with sepiolite and palygorskite surfaces were never studied. Comparing to other minerals, the present values determined for palygorskite were higher than those reported for calcined kaolinitic clays and hydroxyapatites (3–4 kJ mol^−^^1^) [22,23]. As mentioned above, the Lewis acid–base probes more typically used in IGC could not be measured due to their strong interactions with the present samples, which means that the sepiolite and palygorskite surfaces under study have a strong amphoteric character. Other authors have already evaluated the acid–base properties of these two types of clay minerals by IGC, by injecting the more common Lewis acid–base probes and measuring their interactions, whenever possible. In the case of sepiolite, both acidic [10,11] and basic [12] characters have already been reported. For palygorskite, Boudriche et al. (2010) mentioned a strongly acidic character because of the fact that they could not elute basic probes (due to very strong interactions with the palygorskite surface) [13].

The cycloalkane probes (cyclo 6 and cyclo 8) showed significantly lower retention times than their corresponding *n*-alkanes (*n*-hexane and *n*-octane), and, accordingly, the specific interactions of these probes were always negative for all samples (Figure 3 and Figure 4). This fact can be explained by steric hindrance effects of cycloalkanes which limit their access to the structural channels of sepiolite and palygorskite when compared with *n*-alkanes [25]. The additional two carbon atoms in the cyclooctane ring, compared to cyclohexane, confer a higher volume to this probe, and, consequently, its relative access to the channels is more limited (retention time of cyclooctane vs. *n*-octane is speeded up compared to retention time of cyclohexane vs. *n*-hexane), providing even more negative specific interactions than cyclohexane. The specific interaction values of both cycloalkanes with the studied samples increased in the following order: Sep. 2 < Sep. 1 << Pal. This trend is in agreement with the smaller dimensions of the structural channels of palygorskite and with the different pretreatments applied to the sepiolite samples. Finally, the 2,2,4-TMP kept the same behavior as that observed for the cycloalkanes (Figure 3). However, the −ΔG_a_^s^ values, which were obtained by difference between the RTln(V_n_) values of 2,2,4-TMP and *n*-octane, were even more negative due to the branched structure of the chain of this probe.

### 3.4. Nanomorphology Index (${IM\chi }_{T}$)

In order to assess the surface nanoroughness of the studied clay minerals, the nanomorphology indices ${(\mathrm{IM}\chi }_{T})$ were determined based on the measurement of retention times of cycloalkanes (cyclo 6 and cyclo 8) and of a branched alkane (2,2,4-TMP), according to two different approaches. The results are shown in Table 2.

The two different approaches used for the determination of ${\mathrm{IM}\chi }_{T}$ led to the same trends: more negative values of ${\mathrm{IM}\chi }_{T}$ calculated by the Brendlé and Papirer approach correspond to values closer to zero when calculated by the Balard approach, that is, a material with a rougher surface. Based on the results presented in Table 2, it can then be observed that Sep. 2 had the most negative values (in the Brendlé and Papirer approach) and the ones closest to zero (in the Balard approach) for each probe assessed, meaning that Sep. 2 had the roughest surface of all samples. However, the differences from the Sep. 1 sample were not high. On the contrary, Pal showed, by a large difference, the surface with the lowest roughness. From this trend, it was concluded that a higher surface nanoroughness correlated to a higher dispersive component of the surface free energy. This observation was also reported previously by Saada et al. (1995) in the evaluation of the surface properties of illites and kaolinites by IGC [26] and for calcined kaolinitic clays [22]. Rougher surfaces with increasing structural surface defects at the molecular/nanometric scale provide a better insertion of the *n*-alkanes into the material structure. Boudriche et al. (2010) also determined the nanomorphology indices (Balard approach) for a palygorskite sample with cyclooctane and 2,3,4-TMP and obtained ${\mathrm{IM}\chi }_{T}$ values of 0.22 and 0.69, respectively [13]. Comparing the ${\mathrm{IM}\chi }_{T}$ values of cyclooctane, it can be said that our palygorskite sample showed a flatter surface.

## 4. Commercial Applications of the Mineral Samples and Relevance of IGC Results

At present time, regarding the studied samples, sepiolite 1 is used as a sorbent and carrier for chemicals; to control the flow of some fluid systems, such as bitumen sheets, particularly at higher temperatures; and as semireinforcing mineral filler for rubber and polar polymers. Sepiolite 2 is used to provide a thixotropic and pseudoplastic behavior to water-based paints and coatings; as a suspending agent to avoid settling of pigments and fillers; and as a reinforcing filler for polar polymers. When surface-modified, sepiolite 2 can be incorporated in different polymer compounds to improve mechanical properties and fire resistance. Palygorskite is also used as a rheological additive, particularly for building products, such as mortars, wet and dry joint compounds, adhesives, and sealants. It provides a shear-thinning rheological behavior; improves pumpability, workability, and spreading of the compositions; and improves sag resistance when applied in vertical walls.

The present results obtained by IGC demonstrate the high potential of the studied mineral samples for the compatibilization with polymeric matrices with high polarity (e.g., polysaccharide and polyacrylate matrices) and adsorption of a wide variety of substances present in liquid and gaseous effluent streams (e.g., formaldehyde, chlorine- and sulfur-containing compounds).

## 5. Conclusions

In this work, two sepiolite samples, with different pretreatments, and one palygorskite sample were extensively characterized by inverse gas chromatography (IGC) and compared. Based on a previous analysis by X-ray diffraction, the two sepiolite samples were found to have a similar mineralogical composition, as expected, with the presence of only sepiolite. On the other hand, the palygorskite sample was more heterogeneous, composed not only of palygorskite but also of various contaminants such as quartz.

The results obtained by IGC at 240 °C, after previous preconditioning of all samples for the removal of all zeolitic water, revealed that sepiolite samples had higher γ_s_^d^ values than palygorskite, which was related to the larger dimensions of the tunnels and channels present in the structure of the sepiolite vs. palygorskite. It was also observed that the γ_s_^d^ values of sepiolites were slightly affected by the different applied pretreatments. The nonelution of typical Lewis basic, amphoteric, or acidic probes revealed very strong interactions of those probes with the studied minerals and allowed us to conclude that the analyzed sepiolite and palygorskite surfaces have a very strong amphoteric character. On the other hand, the injections of cyclohexane, cyclooctane, and 2,2,4-trimethylpentane showed that steric hindrance of these cyclic and branched alkanes makes it difficult for them to reach the structural channels of the minerals. The nanomorphology indices ${(\mathrm{IM}\chi }_{T})$, calculated based on the retention times of the latter set of probes, showed that the sepiolites clearly exhibited a higher nanoroughness compared to the palygorskite sample, which behaved more like a flatter surface.

It should be highlighted that the very high values of surface energy and the developed amphoteric character observed for the sepiolite and palygorskite samples under study, together with their high specific surface area, even compared with other samples of these minerals previously reported, may favor adsorption processes with volatile organic compounds or matrix–filler interactions regarding the production of composite structures with matrices able to establish Lewis acid–base interactions.

## Figures and Tables

**Figure 1 nanomaterials-11-01579-f001:**
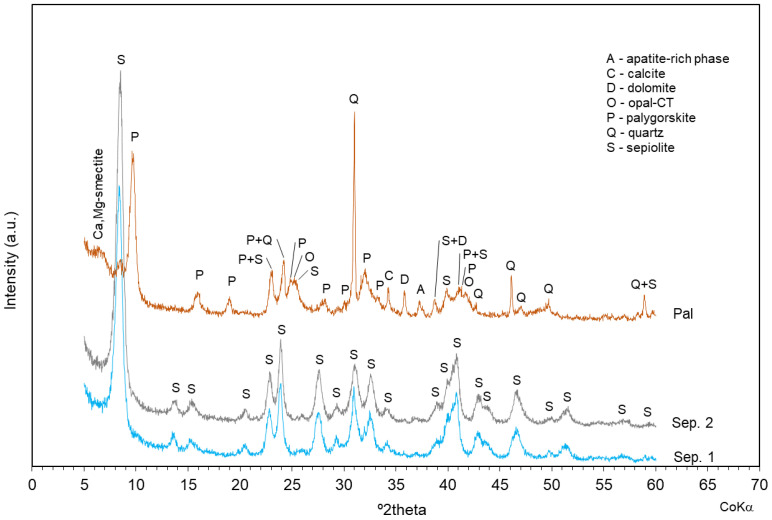
Mineralogical characterization of sepiolite and palygorskite samples by X-ray diffraction.

**Figure 2 nanomaterials-11-01579-f002:**
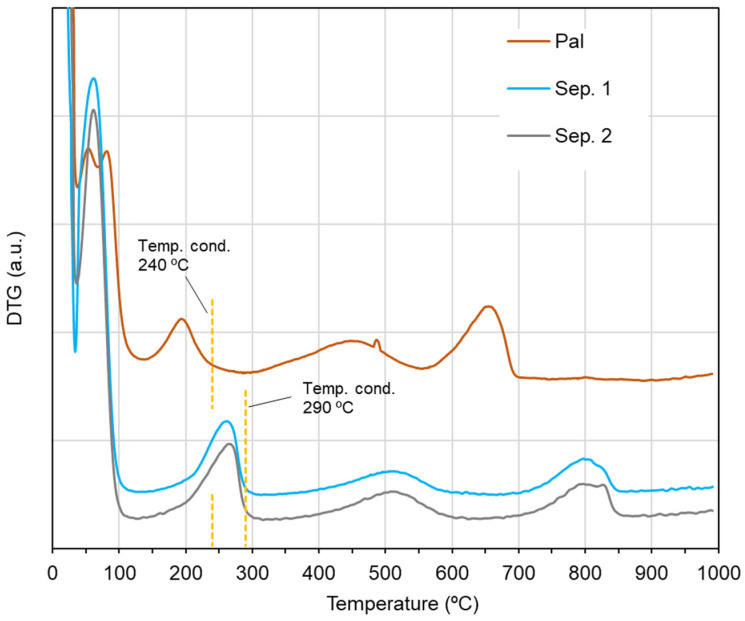
Derivative of the thermogravimetry (DTG) for the sepiolite and palygorskite samples.

**Figure 3 nanomaterials-11-01579-f003:**
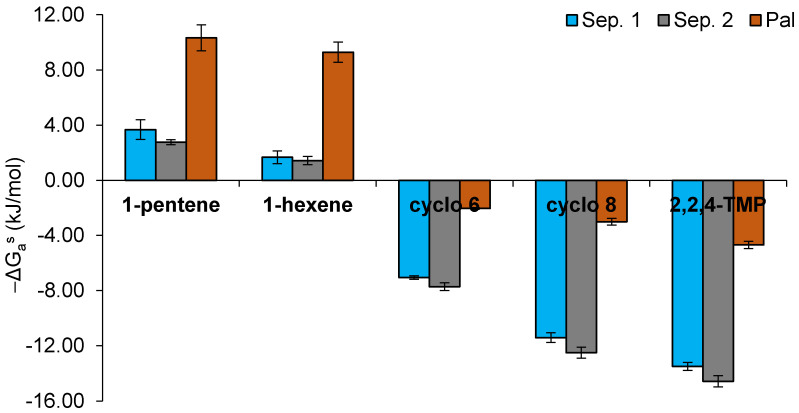
Specific interaction parameter (−ΔG_a_^s^, kJ mol^−1^) for the adsorption of 1-alkenes (1-pentene and 1-hexene), cycloalkanes (cyclohexane (cyclo 6) and cyclooctane (cyclo 8)), and a branched alkane (2,2,4-trimethylpentane (2,2,4-TMP)) on the surface of sepiolite and palygorskite samples at 240 °C.

**Figure 4 nanomaterials-11-01579-f004:**
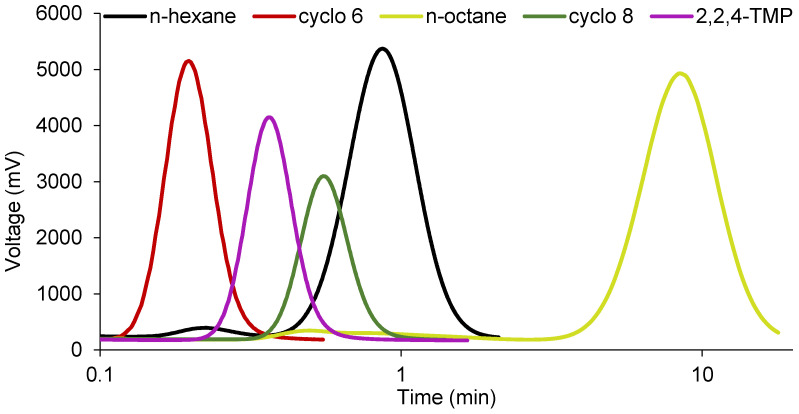
Examples of chromatograms for the elution of *n*-hexane, cyclohexane (cyclo 6), *n*-octane, cyclooctane (cyclo 8), and 2,2,4-trimethylpentane (2,2,4-TMP) in sepiolite 2 at 240 °C (flow rate of 63 mL min^−^^1^). Due to large differences between probes in the values of retention times, these are shown on a logarithmic scale for a better visualization.

**Table 1 nanomaterials-11-01579-t001:** Dispersive component of the surface free energy of sepiolite and palygorskite samples at 240 °C.

Sample	γ_s_^d^ (Schultz–Lavielle) (mJ m^−2^)	γ_s_^d^ (Dorris–Gray) (mJ m^−2^)
Sep. 1	135 ± 11	205 ± 16
Sep. 2	151 ± 4	229 ± 6
Pal	110 ± 5	166 ± 8

**Table 2 nanomaterials-11-01579-t002:** Nanomorphology indices $\left({\mathrm{IM}\chi }_{T}\right)$, as calculated by two methods, of sepiolite and palygorskite samples at 240 °C.

Probes	${\mathrm{IM}\chi }_{T}$ (%, Brendlé and Papirer, 1997)			${\mathrm{IM}\chi }_{T}$ (Balard et al. 2000)		
	Sep. 1	Sep. 2	Pal	Sep. 1	Sep. 2	Pal
Cyclo 6	−24.1 ± 1.7	−25.7 ± 0.4	−9.6 ± 0.4	0.16 ± 0.010	0.15 ± 0.003	0.53 ± 0.010
Cyclo 8	−30.0 ± 0.6	−32.9 ± 0.5	−11.7 ± 0.5	0.05 ± 0.005	0.04 ± 0.003	0.35 ± 0.013
2,2,4-TMP	−26.6 ± 1.2	−30.0 ± 0.5	−5.6 ± 0.7	0.09 ± 0.002	0.07 ± 0.005	0.64 ± 0.030

## Data Availability

The data presented in this study are available on request from the corresponding authors. The data are not publicly available due to lack of adequate repository.

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
