# Peer review of "Comparison of Surface Properties of Sepiolite and Palygorskite: Surface Energy and Nanoroughness"

_nanomaterials, 2021, doi:10.3390/nano11061579_

Round 1

Reviewer 1 Report

Some questions need to be clarified. 

  1. Page 5, XRD can examine the mineral phases (not mineral composition)
  2. The objective/motivation is not strong to support this MS.
  3. How is the difference between sepiolite and palygorskite ? I think this should be emphasized in the conclusion or some suitable place in the content.
  4. How to apply these two minerals (for the practical applications), it needs to describe in the context.
  5. The authors should add the ratio of mineral phases for these two minerals because this would affect their adsorption ability.  

Author Response

Thank you very much for your comments. The responses follow below.

 Some questions need to be clarified. 

Page 5, XRD can examine the mineral phases (not mineral composition).

Response: The reviewer is right. It was not evaluated the composition of the mineral phases by XRD, only the mineral phases present. We have changed “The sepiolite samples (Sep. 1 and Sep. 2) and the palygorskite sample (Pal) were previously analyzed by X-ray diffraction to evaluate their mineralogical composition” to “The sepiolite samples (Sep. 1 and Sep. 2) and the palygorskite sample (Pal) were previously analyzed by X-ray diffraction to evaluate the mineral phases present”.

The objective/motivation is not strong to support this MS.

Response: It was said in the final paragraph of the introduction section “The aim of the present work was to make a comparative study of two sepiolite samples and one palygorskite sample (provided by TOLSA), measured in the same temperature conditions and after an appropriate preconditioning to remove the zeolitic water”. This underlying idea is also present in the title, “Comparison of surface properties of sepiolite and palygorskite: Surface energy and nanoroughness”. The obtained results could have interest for optimizing adsorption processes, or in the area of the production of composites, as stated in the last paragraph of the conclusions: “It should be highlighted that the very high values of surface energy and the developed amphoteric character observed for the sepiolite and palygorskite samples under study, together with their high specific surface area… may favor adsorption processes with volatile organic compounds or matrix-filler interactions regarding the production of composite structures with matrices able to establish Lewis acid-base interactions”. We have now tried to enhance the objectives of the present work in the text. For that, we added a sentence in the end of the introduction section, “The obtained results could be of valuable interest for optimizing adsorption processes using this type of clay minerals, or in the area of the production of composites, where the accessed parameters can be used to predict compatibility between the clay surface and other components”.

How is the difference between sepiolite and palygorskite? I think this should be emphasized in the conclusion or some suitable place in the content.

Response: Structurally speaking, the difference between sepiolite and palygorskite has been described in the first paragraph of the introduction section. “Sepiolite and palygorskite are fibrous clay minerals composed by discontinuous structural blocks, alternating with structural tunnels… Sepiolite is a trioctahedral mineral with the ideal formula Si12O30Mg8(OH)4(OH2)4.8H2O, having all the octahedral sites occupied by magnesium. In contrast, palygorskite, with the ideal formula Si8O20(Al2Mg2)(OH)2(OH2)4.4H2O, has a dioctahedral character and only four of the five octahedral sites are occupied, mainly with magnesium and aluminum ions [2,3]. The structural tunnels of these two minerals have cross-section dimensions of 1.06 × 0.37 nm2 and 0.64 × 0.37 nm2 for sepiolite and palygorskite, respectively, and inside them, there are exchangeable cations and zeolitic water.” If the reviewer is referring to the IGC results obtained for sepiolite and palygorskite samples, the differences in the results for the two minerals have been highlighted throughout all the document, including in the conclusions section (see all the second paragraph of the conclusions).

How to apply these two minerals (for the practical applications), it needs to describe in the context.

Response: Practical applications (e.g., adsorptive processes and composite production) had been already mentioned briefly in the last paragraph of the conclusion section. However, we agree with the reviewer that this issue could be better described in the context. We added now a new section in the manuscript before the conclusions describing some commercial applications of the samples used in this study and interest of the IGC measurements, “Commercial applications of the mineral samples and relevance of IGC results”.

The authors should add the ratio of mineral phases for these two minerals because this would affect their adsorption ability. 

Response: The detailed characterization of the sample minerals used in this study was already described in references 14 (for sepiolite samples) and 15 (for palygorskite sample), as mentioned in section 2.1 (third paragraph): “The general characterization of these fibrous clay samples was reported in detail elsewhere [14,15]”.

According to the X-ray diffraction data, the sepiolite samples only show sepiolite. The palygorskite sample is more complex and shows other mineral phases besides palygorskite, being quartz the most dominant second phase.

We had no way to obtain the ratio of the several mineral phases in the studied palygorskite sample. As the mineral phases found in the palygorskite sample share common elements (for instance Si is present in palygorskite but also in quartz and sepiolite), assessing the quantitative composition of the mineral phases in the sample by stoichiometry using X-ray fluorescence data is not feasible.

Reviewer 2 Report

Criticism

The descriptions of the sample preparation and experimental methods are mostly appropriate. However, the quality/purity of the carrier gas (He) used in IGC measurements was not given. Furthermore, the accuracy and the precision of the retention time measurements are not reported.

The authors did not explained, how were calculated the specific retention volumes from the retention time data, how were measured the input/output column pressures and the carrier gas flow and temperature, and what were the accuracies and the precisions of these measurements.

The paper mixed SI and other units (e.g. °C) but used the same symbol (T) for both quantities.

Author Response

Thank you very much for your comments. The responses follow below.

The descriptions of the sample preparation and experimental methods are mostly appropriate. However, the quality/purity of the carrier gas (He) used in IGC measurements was not given. Furthermore, the accuracy and the precision of the retention time measurements are not reported.

Response: As for the quality/purity of the carrier gas (He) used in IGC measurements, this information was added in the experimental section (section 2.2). As for the precision of the retention time measurements, variations between different injections of the same probe were not higher than 5%; this information was also added in the manuscript in section 2.2.

The authors did not explain, how were calculated the specific retention volumes from the retention time data, how were measured the input/output column pressures and the carrier gas flow and temperature, and what were the accuracies and the precisions of these measurements.

Response: Retention volumes were calculated following procedures commonly used for IGC. In particular, Vn = (tr-t0). F. J, where Vn is the net retention volume, tr is the retention time of the probe, to is the retention time of the non-interacting probe (methane), F is the flow rate of the inert carrier gas (measured with a digital flow meter), and J is the James-Martin compression correction factor. In the text, we have said in section 2.3, “The principles and the mathematical aspects of the IGC theory at infinite dilution conditions have been widely described in literature [17,6-8]”. Thus, we think it is not required to add more information on this issue, which can be find in any paper reporting some basic IGC theory.

Inlet pressure was pre-programmed in the gas chromatograph equipment for each analysis, using digital pressure control (±0.01 bar). Outlet pressure was assumed to be equal to the atmospheric pressure. Temperatures (oven, injector, and detector) were also pre-programmed in the gas chromatograph (±1 ºC).

The carrier gas flow rate was separately measured using a digital flow meter (±0.1 ml/min). This information was added to the manuscript in section 2.2.

The paper mixed SI and other units (e.g. °C) but used the same symbol (T) for both quantities.

Response:

Temperature has been expressed in degree Celsius, in all the text. Only for the calculations using the formalism of thermodynamics the temperature had to be expressed in Kelvin, and in this case, we speak of absolute temperature, as stated in the experimental section (line 173). In Figure 2, correction was made from “T cond.” to “Temp. cond.”

Round 2

Reviewer 2 Report

The Authors presented in their answers the method of the calculation of retention volume:

„Vn = (tr-t0). F. J, where Vis the net retention volume, tr is the retention time of the probe, to is the retention time of the non-interacting probe (methane), F is the flow rate of the inert carrier gas (measured with a digital flow meter), and J is the James-Martin compression correction factor.”

However, for the Vn calculation one needs not only the pressure correction (James factor), but the temperature correction, too. Because the flow rate (F) is mostly measured at room temperature, outside of the IGC column, therefore a temperature correction is absolutely needed to IGC column temperature, especially, if the IGC column temperature is much higher than the temperature of the flow rate measurement.

The lack of the temperature correction makes the presented surface energy data at 240 C questionable.

Author Response

However, for the Vn calculation one needs not only the pressure correction (James factor), but the temperature correction, too. Because the flow rate (F) is mostly measured at room temperature, outside of the IGC column, therefore a temperature correction is absolutely needed to IGC column temperature, especially, if the IGC column temperature is much higher than the temperature of the flow rate measurement.

The lack of the temperature correction makes the presented surface energy data at 240 C questionable.

Response: Thank you very much for the comment. Indeed, the flow rate of the carrier gas, for a more adequate determination of the retention volume of the probes, needs a correction of temperature. However, this correction will be a constant value at a certain working temperature in the column, and does not affect the dispersive component of the surface energy, the specific interaction parameters measured or the nanomorphology index values obtained of a specific material. That is, the correction in flow rate will be the same for all probes, and the dispersive component of the surface energy, as the slope of RTln(Vn) as a function of 2N.ald)0.5  will be exactly the same. The specific interactions of adsorption (-ΔGas) of the injected probes on the surface of the studied materials, as the difference between the RTln(Vn) values of 1-alkenes, cycloalkanes or branched alkane and the RTln(Vn) values of the corresponding n-alkanes with the same number of carbon atoms will be the same as well:  RTln(Vn.(alkene).K)- RTln(Vn.(alkane).K)= RTln((Vn (alkene).K/(Vn.(alkane).K))= RTln((Vn (alkene)/(Vn.(alkane)) and the same applied for nanoroughness. Overall, none of the parameters presented in the paper will be affected by a new correction applied to the calculation of retention volumes.

Round 3

Reviewer 2 Report

The Authors stated in their 2nd response that the temperature „correction will be a constant value at a certain working temperature in the column, and does not affect the dispersive component of the surface energy” and „That is, the correction in flow rate will be the same for all probes”.

If these statements would be right, only the retention time (tR) measurements would be enough for surface energy calculation, because all  the other correction factors (temperature, flow rate, etc) must be the same during the experiments, therefore they do not affect the slope of the ln(tR) in a logarithm scale. Unfortunately, in a real GC system the flow rates of carrier gases, the measuring temperatures in the oven and in the lab, and the atmospheric pressure varies from day-after-day. Therefore the surface energy calculations in the literature are based on the Vg (Vn) values and not on the tR data.

If all the corrections are constant and are “the same for all probes” as it is stated by the Authors, then the Authors must have got a wonder-GC!